# MCP-R1: Generalized Real-World Task Agent Mastering Dozens of Tools

## Abstract

Modern agentic models require strong capabilities for orchestrating external tools to interact with complex environments. However, existing tool-integration approaches support only a narrow range of tools and lack a unified calling standard. Consequently, they devote little attention to real-world tasks and struggle to transfer to unseen tools. The emergence of the Model Context protocol (MCP) presents an open standard for two-way connections between external tools and agents. To this end, we introduce MCP-R1, a new paradigm designed to enhance models' universal tool-interaction capabilities. We first construct a virtual-real integrated MCP tool system, supporting 17 MCP servers with 60+ tools, each sourced from real-world services to ensure diversity and authenticity during training. Based on the tool system, we further propose a scalable pipeline for generating multi-tool invocation data. In addition, going beyond rule-based rewards commonly used in QA tasks, we introduce a trajectory-based reward mechanism to evaluate the agent's performance in goal-driven tasks. Thanks to the unified tool-interaction standard and our training pipeline, MCP-R1 has generic interacting ability across a broad set of tools, demonstrates strong performance on practical tasks across diverse scenarios, while flexibly adapting to unseen tools. Our experiments span several challenging domains including search (GAIA, WebWalker), general tool calling (MCP-Universe), and practical task execution. The strong performance of MCP-R1 underscores the effectiveness of our training paradigm, offering valuable insights and a scalable approach for developing general agentic models.

## 1 Introduction

Agentic models with the critical capability of interacting with environments via external tools have rapidly evolved into general-purpose task solvers. With the two-way tool-interaction, the model can not only acquire information beyond parameter memory from the environment, but also execute operations to change the environment state according to user query, thereby significantly expanding the boundaries of the agentic model's capabilities. Recently, tool invocation, dominated by code and search engines, has garnered attention as a foundational capability for agentic models (Li et al., 2025d; Singh et al., 2025; Zheng et al., 2025; Dong et al., 2025b;a; Gao et al., 2025; Jin et al., 2025). Although significant progress has been made in their respective tasks, training models to proficiently utilize dozens of tools across a broader range of practical tasks remains an unresolved challenge. The Model Context Protocol (MCP), introduced by Anthropic (Anthropic, 2024), provides a unified standard for AI systems to interact with external tools, gaining rapid traction (OpenAI; Google; Cursor) and laying the groundwork for general-purpose tool interaction.

Currently, some work has successfully introduced code/search engines to enhance the model's capabilities in math/retrieval tasks. WebThinker (Li et al., 2025c) trains models to dynamically search, navigate, and extract information from websites. ToRL (Li et al., 2025d) and ReTool (Feng et al., 2025) focus on mathematical reasoning, training models through RL to flexibly utilize code tools for assisted computation. Furthermore, some works begin to explore the impact of introducing external tools on the model generation process. For instance, ARPO (Dong et al., 2025b) observes an increase in the entropy of generated tokens following tool invocation, and designs an adaptive rollout mechanism to facilitate exploration with uncertainty.

Despite recent advances, existing approaches still suffer from several problems. *(1) Limited tool coverage:* Currently, most tool integration approaches only focus on how to incorporate a single tool to enhance model performance on a single task category. Furthermore, these studies have focused solely on answer-driven tasks where precise answers can be obtained, neglecting goal-driven tasks that require actions to alter environmental states rather than providing answers, which is prevalent in real-world scenarios. *(2) Lack of unified interaction standards:* Existing approaches rely on fragmented, customized tools that trap language models in isolated information silos. The lack of unified interaction standards prevents models from transferring prior knowledge accumulated on seen tools to unseen tools, directly limiting their generic interacting ability across a broad set of tools. The introduction of the MCP protocol has, to some extent, facilitated closed-source model services, but open-source agentic model training has lacked attention to unified tool standards like MCP, resulting in fragmented, task-specific tool formats that hinder cross-schema adaptability and limit generalization in tool use.

To address these challenges, we introduce MCP-R1, an agentic training framework designed to enhance the model's general-purpose tool-interaction capabilities. To improve the ability of tool-interaction with real-world environments, we developed a ***Virtual-Real Integrated MCP Tool System***, which supports approximately 17 MCP Servers and over 60 tools. Each tool is sourced from real-world services and utilizes MCP as a standardized interface for interaction. Based on the constructed Tool System, we further introduce the ***Scalable Data-generation Pipeline*** to generate multi-tool invocation tasks. This pipeline encompasses both answer-driven tasks, which retrieve static standard answers, and goal-driven tasks, which dynamically interact with the environment and execute operations to alter its state. For more complex goal-driven tasks lacking predefined solutions, we introduce a trajectory-based evaluation method. By assessing the model's trajectory against a comprehensive, predefined rubric, this approach unlocks the potential to optimize agents' intricate interaction processes. In summary, the key contributions can be described as follows:

- We present MCP-R1, a paradigm that integrates tools, data synthesis, and training approaches to enhance the general tool-use capabilities of models. Moreover, we move beyond traditional answer-driven tasks, empowering models to tackle real-world problems.

- We develop a Virtual-Real Integrated MCP Tool System with 60+ real-world tools compliant with the MCP standard as a standardized training environment, together with a Scalable Data-generation Pipeline to generate diverse multi-tool invocation data, both answer-driven tasks and goal-driven tasks.

- Our experiments demonstrate that MCP-R1 exhibits powerful performance across multiple challenging domains (specific & general) and diverse task types (answer-driven & goal-driven). This provides practical insights for exploring general-purpose tool interaction capabilities within agentic models.

## 2 RELATED WORKS

### 2.1 TOOL-INTEGRATED MODELS

Large Language Models (LLMs) (Gemini et al., 2023; Team et al., 2025a; Zeng et al., 2025) are inherently constrained by their static training data, resulting in knowledge cutoffs and cannot interact with external systems. To address these issues, several works try to augment LLMs with tools, enabling them to interact with environments by using external APIs and services. ReAct (Yao et al., 2022) first showed how models can combine reasoning and actions. Later, frameworks like ART (Paranjape et al., 2023; Sun et al., 2025) built on this idea, allowing models to automatically choose tools and solve multi-step tasks. Building on this foundation, subsequent work has focused on scaling and optimizing tool-use. ToolLLM (Qin et al., 2024) collects extensive APIs and employs ChatGPT to generate trajectories for instruction fine-tuning. Search-o1 (Li et al., 2025b) introduces the autonomous search mechanism via prompt-guided, thereby enhancing performance on complex reasoning tasks. Recently, to unify the tool interaction format, Anthropic introduced MCP (Anthropic, 2024), which defines a unified interface for tool implementations, gradually gaining favor among AI service providers (OpenAI; Cline). Building upon this foundation, we incorporate it into MCP-R1, enabling MCP-R1 to adapt to a wide range of existing tools and generalize to unseen tools. Unlike prior systems such as T3-Agent (Gao et al., 2024), which focus on multimodal QA-style tool

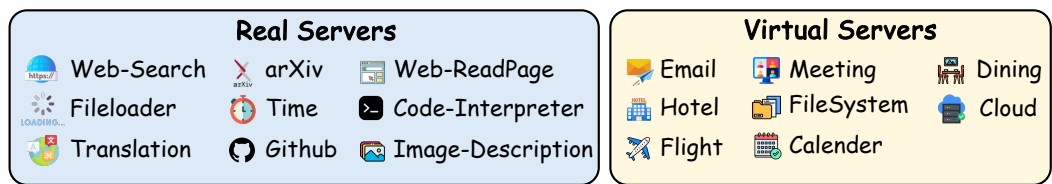

Figure 1: List of MCP servers in the Virtual-Real Integrated MCP Tool System.

use under proprietary formats, our work targets multi-step, goal-driven agent orchestration with a unified interaction standard and a scalable data-generation pipeline that enables autonomous exploration and stronger generalization in real-world tool-use tasks. Compared with API-based dataset generation approaches such as ToolFormer (Schick et al., 2023), our method introduces a unified interaction protocol, multi-turn goal-oriented data construction, and an SFT-RL training paradigm that together enable stronger cross-tool generalization and long-horizon task competence beyond single-step API invocation.

### 2.2 AGENTIC REINFORCEMENT LEARNING

Agents have long been a research hotspot due to their ability to execute practical tasks in specific scenarios (Team et al., 2025b). With the rise of Reinforcement Learning (RL), some works have begun exploring the use of RL to train agents' interaction capabilities in dynamic environments. (Lù et al., 2025; Shridhar et al., 2020; Mialon et al., 2024). Some approaches use trajectory-level rewards to refine an agent's ability to integrate reasoning with environmental interactions (Wang et al., 2025; Zhou et al., 2024; Shang et al., 2025). Frameworks such as ToolRL (Qian et al., 2025), ReTool (Feng et al., 2025), and Tool-Star (Dong et al., 2025a) train models to leverage search engines or Python compilers to solve complex problems. WebDancer (Wu et al., 2025a) and WebSailor (Li et al., 2025a) instead employ a meticulously designed extensible data construction pipeline, coupled with SFT-RL two-stage training, to achieve outstanding performance in the search domain. ARPO (Dong et al., 2025b) focuses on the impact of introducing external tools on the model generation process, proposing an entropy-based adaptive rollout mechanism to promote diverse trajectory exploration in tool-invocation scenarios. In this work, we regard tool interaction as a fundamental capability of the agentic model and attempt to stimulate the model's tool invocation for general-purpose tasks through the Virtual-Real Integrated MCP Tool System with 60+ real-world tools.

## 3 MCP-R1: GENERALIZED REAL-WORLD TASK AGENT

### 3.1 TOOL SYNTHESIS AND DATA CONSTRUCTION

Existing tool-integration approaches (Li et al., 2025c; Dong et al., 2025a; Feng et al., 2025) predominantly focus on leveraging search and code execution tools to solve problems within the domains of web navigation, programming, and so on. However, these approaches exhibit poor generalization when applied to a broader spectrum of real-world tasks. To improve the generalization capability of agentic models and enhance their ability to interact with the real world, we first develop a **Virtual-Real Integrated MCP Tool System**. By employing a unified MCP protocol, this system provides the model with a standardized interface for interacting with real-world tasks. Based on the tool system, a **Scalable Data-generation Pipeline** is further introduced, which is capable of generating diverse tasks. These tasks span from answer-driven problems with definite answers to complex goal-driven tasks that are resolved through intricate environmental interactions.

#### 3.1.1 VIRTUAL-REAL INTEGRATED MCP TOOL SYSTEM

While real-world MCP servers provide high-fidelity environments for agent-environment interaction, they come with *(1) significant monetary and temporal costs* (e.g., API call fees and response latency), *(2) considerable security vulnerabilities*, and *(3) limited scalability*. The security risks are particularly notable, as these systems may require user authentication and could potentially execute irreversible operations (*e.g.*, deleting emails). These limitations become especially pronounced

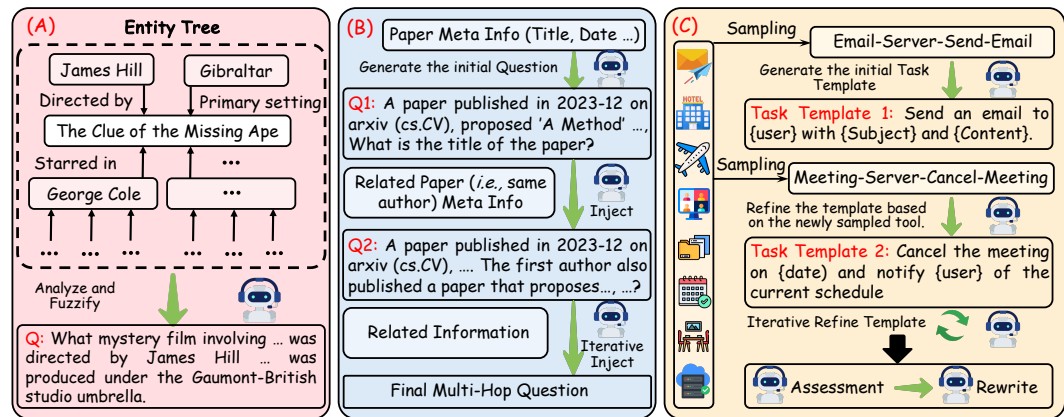

Figure 2: Data Construction Pipeline. The first two figures show the pipeline of the answer-driven task. The third one represents goal-driven tasks. (A) Knowledge tree-based construction. (B) Iterative trajectory-based construction. (C) Iterative, template-based construction.

in the context of large-scale agent training. Conversely, simulated environments offer a robust alternative, distinguished by their economic efficiency, rapid responsiveness, and providing a secure sandbox environment conducive to extensive trial-and-error learning. Therefore, constructing a hybrid tool system that integrates virtual and real-world MCP servers is of paramount importance. This approach not only enhances overall system security and minimizes training costs but does so without compromising the authenticity of the agent's interactive experience. Our MCP server list is shown in Fig. 1 and more details about the tools are provided in our Appendix.

**Real MCP servers** provide an environment that allows agents to interact directly with live and publicly accessible systems. These tools are primarily used in scenarios where agents must interact with dynamic, unpredictable, and difficult-to-simulate information sources. A notable characteristic of real-world tools is that their output is often random and non-reproducible, such as real-time results from internet search engines. We encapsulate these real-world APIs (*e.g.*, Google Search API) in servers that adhere to the MCP standard.

**Virtual MCP servers**, in contrast, are high-fidelity simulations designed to mirror the functionality of their real-world counterparts. They are primarily used in three scenarios: (1) for tasks where the output can be programmatically modeled based on the input; (2) for operations that require user authentication, involve private data, or have strict security guarantees, such as accessing a personal Gmail account; and (3) as a cost-effective alternative to interacting with expensive proprietary APIs. These tools are built by hand-implementing a simulation service that accurately replicates the MCP interface of the real tool. This approach ensures that agents trained specifically in a safe and controllable virtual environment can be seamlessly deployed to run alongside the real tool, achieving zero-shot transfer without further fine-tuning.

### 3.1.2 SCALABLE DATA GENERATION PIPELINE

To systematically construct our dataset, we categorize real-world tasks into two categories. *(1) Answer-driven tasks:* The goal is to retrieve a single, static, and verifiable ground-truth answer, such as using web search for question answering. Most of the existing works (Li et al., 2025a; Dong et al., 2025a) mainly focus on this type of task. *(2) Goal-driven tasks:* These tasks lack a single, predefined solution. Their success depends on effectively achieving a specific goal through a series of decisions and actions. This requires the agent to interact with the external environment (*e.g.*, file system). The agent must be able to observe the environment, formulate plans, execute actions, and iteratively refine its policy based on subsequent feedback.

**Answer-driven Task Construction.** We utilize two approaches to generate the answer-driven task. The first approach, known as **knowledge tree-based construction**, focuses on generating questions that require synthesizing information from multiple sources. As shown in the Fig. 2 (a), this process begins by extracting a sub-knowledge tree from a structured database (*e.g.*, Wikipedia). Each node in

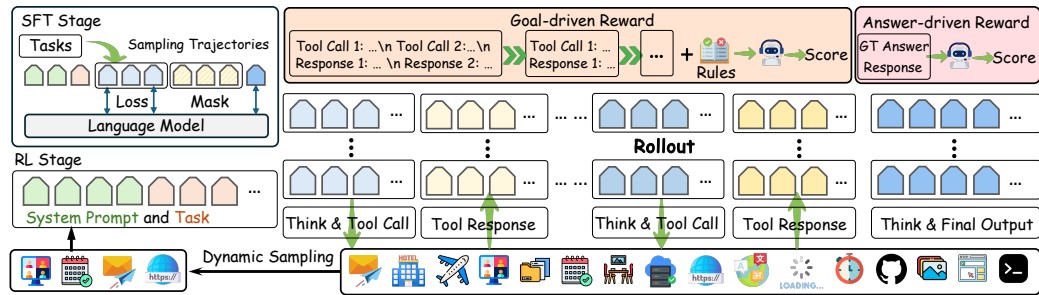

Figure 3: Agentic Training Pipeline of MCP-R1. Our training pipeline includes two stages: (1) Fine-tuning the LLM based on the sampling trajectories and (2) Post-training LLM with the designed reward.

this tree represents an entity, supported by multiple independent sources. A LLM is then prompted to analyze the target node and its associated information, intentionally obscuring the entity to construct a complex question. Such questions cannot be answered with a single query; they require the agent to gather diverse evidence from various sources to ultimately determine the answer. The other approach is **Iterative Trajectory-Based Construction** which is designed to generate multi-hop reasoning tasks. This approach functions by employing a language model to iteratively build a query dependency chain through the continuous injection of new information. In this process, the output of a preceding step is programmatically used as the foundational input for the subsequent query. As shown in Fig. 2 (b), a task might first require an agent to identify a high-order entity, and then leverage that entity to resolve a more specific, related question. Consequently, this structure compels the agent to execute a series of logically interdependent queries to obtain the final answer.

**Goal-driven Task Construction.** To generate goal-driven tasks across multi-level difficulty, we employ an **iterative, template-based strategy** as shown in Fig. 2 (c). The first step is to classify all available servers and their corresponding tools based on the scenario. Tools with similar application contexts, such as `flight-server` and `hotel-server`, are grouped into the same category. Building upon this categorical foundation, we designed an iterative process for task template construction. Initially, a single tool is randomly selected and the LLM is instructed to generate a foundational task template. This template is not a concrete query but rather an abstract framework containing placeholders (*e.g.*, `{departure}`). The process then iterates: a new tool is sampled, with a higher probability of being drawn from the same category, and the LLM is tasked with integrating its functionality into the existing template. This procedure results in a more complex, multi-step task requiring the invocation of multiple tools. By repeating this cycle, we collect a structured hierarchy of task templates, including both single-tool and multi-tool scenarios.

Following the template generation, each template undergoes an assessment process. An LLM first performs an automated assessment of the template's logical feasibility and its solvability using the provided tools and then a final manual review is executed to guarantee the quality of the task. Templates that pass this verification are then instantiated with concrete data, where placeholders are populated with specific details such as user information, times, and locations. Finally, to ensure that tasks are closely related to real-world tasks, we prompt the LLM to inject context into each task. For example, a simple instruction to book a flight and hotel might be transformed into the more realistic instruction to plan a business trip.

## 3.2 AGENT LEARNING WITH DYNAMIC TOOLS

Following Guo et al. (2025), we adopt a two-stage training paradigm based on our constructed dataset. The initial stage consists of Supervised Fine-Tuning (SFT) using the sampled trajectories, followed by a reinforcement learning stage with the designed reward.

### 3.2.1 COLD START WITH SAMPLING TRAJECTORY

We first generate trajectories using the `Qwen3-Max-Preview` (Yang et al., 2025) based on our dataset, adhering to the Qwen tool-calling format. We discard trajectories that are either solved

correctly by the model without tool use or remain incorrectly answered even with tools. Finally, the SFT dataset is composed of 3K trajectories and the model is updated with the following loss:

$$\mathcal{L}_{SFT} = -\sum_{i=1}^{T} \text{mask}(i) \cdot log P_\theta(y_t|y_{<t}, \mathbf{x}), \tag{1}$$

where $\text{mask}(i)$ denotes the mask for the $i$-th token and is set to 0 for any token returned by the tool.

### 3.2.2 Reinforcement Learning

Following the SFT phase, we adopt a RL phase to further enhance the model's capabilities in complex tasks. We employ the GRPO (Shao et al., 2024) algorithm to optimize the model's policy. For different task types (*i.e.*, answer/goal-driven), we utilize distinct strategies to obtain the final reward signal from the LLM judge.

**For answer-driven tasks** that have verifiable ground truth. We leverage the LLM to determine the consistency between the generated answer and the reference solution. **For goal-driven tasks**, which are inherently more complex and lack a single definitive outcome, we develop a trajectory-based evaluation method. For each trajectory, we first extract a structured execution trajectory from the model's output. This trajectory contains not only the specific parameters and return values of each tool call, but also the tool call logic (*e.g.*, the order and concurrency of these calls). Subsequently, the LLM is guided by a comprehensive, predefined rubric to assess the effectiveness of this trajectory. This evaluation rubric examines the logical soundness of each step, the accuracy of the tools invoked, and the degree to which the final result successfully achieves the user's initial goal. This approach transforms the challenge of evaluating performance against vague goals into a structured, systematic evaluation of a series of specific, verifiable actions. Detailed criteria for this evaluation rubric are provided in the Appendix.

In addition, we realize that incorporating the full definitions for all available MCP servers into the model's context consumes significant memory and context window space, degrading training efficiency as the server library expands. To address this issue, we employ a **Dynamic Server Sampling** strategy. For each training sample, we first ensure that all MCP servers required to perform the current task are included in the context (*i.e.*, system prompts). Further, we randomly extract one or two additional MCP servers that are not directly related to the current task from our tool system and inject their definitions into the context. This approach not only ensures that the model can solve the problem with the provided tools but also trains the model to accurately identify and ignore irrelevant ones, thereby enhancing its generalization capabilities.

Finally, we mask the response of tools during RL training and the update process of agentic model can be formulated as follows:

$$\begin{aligned} \mathcal{J}_{\text{RL}}(\theta) =& \mathbb{E}_{q,\{o_i\}} \left[ \frac{1}{G} \sum_{i=1}^{G} \left( \min \left( \frac{\pi_\theta(o_i|q)}{\pi_{\theta_{\text{old}}}(o_i|q)} A_i, \text{clip} \left( \frac{\pi_\theta(o_i|q)}{\pi_{\theta_{\text{old}}}(o_i|q)}, 1-\epsilon, 1+\epsilon \right) A_i \right) \right) \right. \\ & \left. - \beta \mathbb{D}_{\text{KL}}(\pi_\theta \| \pi_{\text{ref}}) \right], \quad \text{where } A_i = \frac{R_i - \text{mean}(\{R_j\})}{\text{std}(R_j)}. \end{aligned} \tag{2}$$

## 4 Experiment

### 4.1 Preliminary

To evaluate the effectiveness of MCP-R1, we conduct experiments on the following benchmarks: (1) **Deep Search:** including General AI Assistant (GAIA) (Mialon et al., 2024) and WebWalkerQA (Wu et al., 2025b). Following previous works (Dong et al., 2025b; Li et al., 2025a), for GAIA, we report the result on the search split. (2) **MCP Benchmarks:** We further evaluate MCP-R1 on MCP-Universe (Luo et al., 2025), an execution-based MCP benchmark specifically designed for evaluating the agent's ability to interact with the external environment by leveraging MCP servers. (3) **Multi-Tool Real-World Task:** We report evaluation results on the full-set of GAIA, with contains real-world questions that require a set of fundamental abilities and generally tool-use proficiency beyond

Table 1: Overall performance on various deep search tasks, including GAIA and WebWalkerQA, with accuracy results for each dataset obtained using llm-as-judge.

| Method | Base Model | GAIA | | | | WebWalkerQA | | | |
|---|---|---|---|---|---|---|---|---|---|
| | | Lv.1 | Lv.2 | Lv.3 | Avg. | Easy | Med. | Hard | Avg. |
| *Direct Reasoning (>=32B)* | | | | | | | | | |
| Qwen3-32B-thinking | - | 26.2 | 12.1 | 0 | 14.9 | 6.9 | 1.1 | 2.9 | 3.1 |
| DeepSeek-R1-32B | - | 21.5 | 13.6 | 0.0 | 14.2 | 7.5 | 1.4 | 4.2 | 3.8 |
| QwQ-32B | - | 30.9 | 6.5 | 5.2 | 18.9 | 7.5 | 2.1 | 4.6 | 4.3 |
| GPT-4o | - | 23.1 | 15.4 | 8.3 | 17.5 | 6.7 | 6.0 | 4.2 | 5.5 |
| DeepSeek-R1-671B | - | 40.5 | 21.2 | 5.2 | 25.2 | 5.0 | 11.8 | 11.3 | 10.0 |
| o4-mini | - | - | - | - | 33.3 | - | - | - | - |
| o1-preview | - | - | - | - | - | 11.9 | 10.4 | 7.9 | 9.9 |
| *Prompt-Based Method* | | | | | | | | | |
| | GPT-4o | 28.2 | 9.6 | 0.0 | 15.5 | 6.7 | 9.5 | 4.2 | 7.0 |
| ReAct | Qwen3-8B | 18.0 | 17.3 | 8.3 | 16.5 | 13.3 | 19.1 | 21.3 | 18.5 |
| | Qwen3-235B | 51.2 | 34.6 | 8.3 | 37.8 | 20.0 | 23.8 | 19.7 | 21.5 |
| Vanilla RAG | Qwen3-8B | 28.2 | 15.4 | 16.7 | 20.4 | 8.9 | 10.7 | 9.9 | 10.0 |
| Search-o1 | Qwen3-8B | 35.9 | 15.4 | 0.0 | 21.4 | 6.7 | 15.5 | 9.7 | 11.5 |
| *RL-based Method* | | | | | | | | | |
| WebThinker | Qwen3-8B | 43.6 | 11.5 | 0.0 | 22.3 | 6.7 | 13.1 | 16.9 | 13.0 |
| WebDancer | Qwen2.5-7B | 41.0 | 30.7 | 0.0 | 31.0 | 40.6 | 44.1 | 28.2 | 36.0 |
| WebSailor | Qwen2.5-7B | - | - | - | 37.9 | - | - | - | - |
| ARPO | Qwen3-8B | 53.9 | 32.7 | 16.7 | 38.8 | 26.7 | 33.3 | 29.6 | 30.5 |
| MCP-R1 | Qwen3-8B | 35.9 | 48.0 | 16.7 | 39.8 | 31.1 | 33.3 | 33.8 | 33.3 |

search engine. We further develop a benchmark, named MCP-RealWorld, to evaluate the task-solving capabilities of the agentic model in real-world scenarios. This benchmark consists of 199 distinct, goal-driven tasks, covering a variety of everyday situations such as travel planning and daily office management. More details can be found in the Appendix.

- **GAIA** (Mialon et al., 2024) is a benchmark for evaluating general AI assistants across reasoning, multimodal understanding, and tool usage. Most questions are text-based, while some include multimodal inputs such as images or spreadsheets and the problems are organized into three levels of difficulty.

- **WebWalkerQA** (Wu et al., 2025b) is a benchmark dataset designed to evaluate the web traversal capabilities of LLMs. The dataset comprises 680 high-quality question-answer pairs spanning more than 1,373 web pages.

- **MCP-Universe** (Luo et al., 2025) is a comprehensive benchmark designed to evaluate large language models in realistic interactions with real-world MCP servers. It spans six core domains: location navigation, repository management, financial analysis, 3D designing, browser automation, and web searching. The benchmark integrates eleven distinct servers and comprises 231 tasks in total.

### 4.2 IMPLEMENTATION DETAILS

For the Supervised Fine-Tuning (SFT) phase, we utilize the Llama Factory framework. We curate a dataset of 3,000 trajectories and set the learning rate to 7e-6. For the Reinforcement Learning (RL) phase, we employed RL-Factory (Chai et al., 2025), an open-source repository based on verl (Sheng et al., 2024) that natively supports the MCP environment. In this phase, we configure the rollout to 4 and adjust the learning rate to 1e-6. We generate 5K data to perform RL training, including 3K answer-driven data and 2K goal-driven data. All experiments were conducted on 8 NVIDIA A800 GPUs. During the evaluation phase, we leverage the Qwen-Agent framework to assess the performance of our models. In our evaluations, we only expose the toolset corresponding to each domain (GAIA, MCP-RealWorld) or the standard toolset provided by the test set (MCP-Universe), rather than giving the model the entire tool inventory.

Table 2: Comparison on MCP-Universe benchmark. Following Luo et al. (2025), we use the ReAct agent pipeline. We report the success rate for each domain and all tasks.

| Model | Size | Location Navigation | Repository Management | Financial Analysis | Browser Automation | Web Searching | Overall Success Rate |
|---|---|---|---|---|---|---|---|
| *Proprietary Models* | | | | | | | |
| Claude-3.7-Sonnet | - | 13.33 | 18.18 | 40.00 | 23.08 | 21.82 | 23.28 |
| Gemini-2.5-Pro | - | 13.33 | 12.12 | 50.00 | 25.64 | 12.73 | 22.76 |
| Gemini-2.5-Flash | - | 15.56 | 12.12 | 37.50 | 30.77 | 14.55 | 22.10 |
| GPT-4.1 | - | 8.89 | 6.06 | 40.00 | 23.08 | 10.91 | 17.79 |
| GPT-4o | - | 8.89 | 9.09 | 35.00 | 12.82 | 9.09 | 14.99 |
| *Open-Source Models* | | | | | | | |
| GLM-4.5 | 355B | 17.78 | 9.09 | 50.00 | 15.38 | 27.27 | 23.90 |
| Kimi-K2 | 1T | 11.11 | 9.09 | 47.50 | 15.38 | 14.55 | 19.53 |
| Qwen3-Coder | 480B | 8.89 | 3.03 | 50.00 | 25.64 | 10.91 | 19.69 |
| Qwen3-235B | 235B | 11.11 | 9.09 | 50.00 | 15.38 | 9.09 | 18.93 |
| DeepSeek-V3 | 671B | 11.11 | 6.06 | 30.00 | 12.82 | 7.27 | 13.45 |
| GPT-OSS-120B | 120B | 6.67 | 6.06 | 35.00 | 5.13 | 5.45 | 11.62 |
| Qwen3-8B | 8B | 4.44 | 0.00 | 17.50 | 2.56 | 3.60 | 5.61 |
| MCP-R1 | 8B | 11.11 | 6.06 | 37.50 | 12.82 | 12.73 | 16.04 |

## 4.3 RESULTS ON DEEP SEARCH.

**Results on Deep Search Benchmarks.** To validate the performance of MCP-R1 in specific complex scenarios, we conducted tests on challenging Deep Search tasks and compared the results against a series of powerful baseline models. As can be seen from Tab. 1, deep search tasks pose significant challenges for existing agent models. Even GPT-4o, lacking tool interaction capabilities, could only achieve scores of 15.5% on GAIA and 7.0% on WebWalker, respectively. By comparison, Qwen-235B achieves scores of 37.8% and 21.5% respectively under identical configurations. This disparity demonstrates the strong limitations of purely prompt-based tool-interaction methods, while the model—as a critical component of agent systems—should prioritise tool-interaction capability as a foundational capability. With only 3K search data involved in RL, MCP-R1 achieved outstanding performance, scoring 39.8% on GAIA and 33.3% on Webwalker, respectively. This significantly surpassed Qwen3-8B under the ReAct framework and yielded the best results among methods of comparable scale. Moreover, MCP-R1 demonstrated comparable or even superior performance to models with significantly larger scales, further validating our approach's efficacy in enhancing model performance on complex tasks through tool interaction. It should be noted that the various baselines presented in Tab. 1 only support one or two specific tools. In Contrast, while achieving outstanding performance on Deep Search, MCP-R1 maintains support for a vast array of tools and retains the capability to execute a broader range of real-world tasks. This enables our approach to stand out among methods limited to a finite set of tools.

**Test-Time Scaling.** Given that agent tasks often exhibit dynamic and multi-round interaction characteristics, we report MCP-R1's pass@K results across various scenarios to more comprehensively demonstrate its potential for utilizing tools in practical tasks. Fig. 4 demonstrates that MCP-R1 exhibits a consistent trend of improvement and scalability across diverse

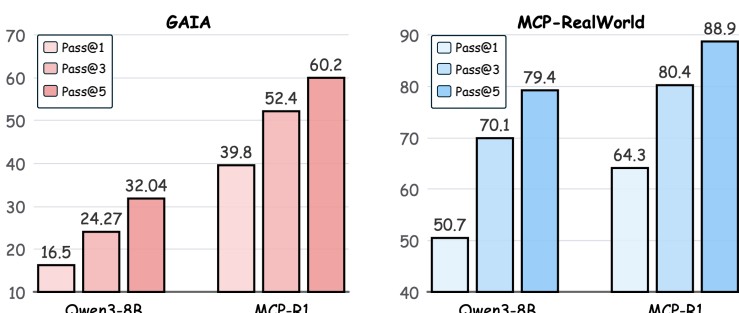

Figure 4: Pass@K results of MCP-R1.

scenarios. Our model achieves remarkable performance on pass@5, particularly scoring 60.2% on GAIA, 88.9% on MCP-RealWorld. Furthermore, compared to the baseline, MCP-R1 exhibits a stronger trend of performance improvement as the number of samples increases. Such stable per-

Table 3: Results of Search split and Entire benchmark on GAIA.

| Method | Split | GAIA | | | |
|--------|-------|------|------|------|------|
| | | Lv.1 | Lv.2 | Lv.3 | Avg. |
| Qwen3-8B | Search | 18.0 | 17.3 | 8.3 | 16.5 |
| Qwen3-8B | All | 22.0 | 23.2 | 4.7 | 17.0 |
| MCP-R1 | Search | 37.8 | 51.0 | 20.0 | 39.8 |
| MCP-R1 | All | 41.5 | 44.2 | 7.7 | 37.6 |

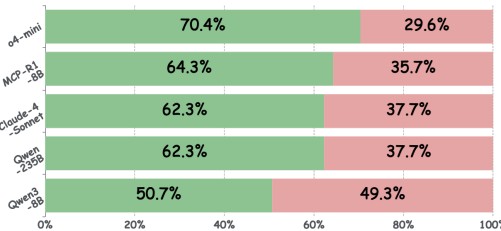

Figure 5: Results on MCP-RealWorld.

formance gains when scaling in samples indicate that through extensive interactions with numerous tools and diverse environments, the model has acquired general tool interaction capabilities beyond specific individual tools. This enables the model to learn how to explore more effectively within the tool-involved sampling space, thereby achieving simultaneous benefits in inference performance and sampling diversity.

## 4.4 RESULTS ON MCP BENCHMARKS.

To further verify the generalization and general-purpose tool-use capability of MCP-R1, we evaluate on MCP-Universe (Luo et al., 2025) benchmark. Specifically, our evaluation is performed on five core subsets within MCP-Universe (*i.e.*, excluding 3D Designing due to the environment issue of Blender). As shown in Tab. 2, MCP-R1 can significantly outperform the baseline model (*i.e.*, Qwen3-8B) across all tasks with an average +10.4% improvement. Besides, the overall performance is comparable to the models with much more capacities (*e.g.*, open-source models larger than 100B). Moreover, MCP-R1 demonstrates remarkable generalization in tool-use scenarios. It is worth noting that most of the servers used in MCP-Universe ( *e.g.* `financial-server`, `google-map-server`) are not included in MCP-R1's training data. Nevertheless, by leveraging the designed scalable data generation pipeline and agent training framework, MCP-R1 is able to effectively utilize these unseen MCP servers, with its performance even surpassing larger models. For example, we surpass DeepSeek-V3 (Liu et al., 2024) with an average +2.59% improvement. These findings indicate that training with the MCP-R1 paradigm on hybrid data is a highly effective strategy for enhancing the universal tool-use capability of agentic models.

## 4.5 RESULTS ON MULTI-TOOL REAL-WORLD TASK.

**Results on GAIA Full-set.** To investigate the performance of MCP-R1 in specific application scenarios, we report evaluation results on the full-set of the General AI Assistant Benchmark (GAIA) (Mialon et al., 2024). The results in Tab. 3 demonstrate that even when applied to the GAIA full-set requiring nearly ten tools, MCP-R1 also achieved an outstanding score of 37.6%, proving its promising capabilities in solving real-world problems. It should be noted that existing tool invocation methods typically support only a limited range of tool categories. Moreover, the customised implementation of these tools prevents them from adapting to usneen's tools via unified interaction standards, thereby failing to meet the practical application requirements of real-world problems. In contrast, MCP-R1, trained with 60+ tools involved under the unified MCP interact standard, inherently covers most real-world scenarios with its capability set and easily adapts to newly introduced tools, demonstrating exceptional versatility and scalability.

**Results on MCP-RealWorld.** To evaluate the real-world task-solving capability of MCP-R1, we further conduct experiments on the self-constructed MCP-RealWorld bench as shown in Fig. 5. It can be seen that MCP-R1 not only surpasses the baseline by a significant margin (*i.e.*, +13.6% improvement) but also outperforms advanced models like Qwen3-235B-A22B (*i.e.*, +2%). Notably, the performance of MCP-R1 exceeds Claude-4-Sonnet-Thinking on our benchmarks, which is because our method can effectively reduce the irreversible operations caused by flawed tool-calling sequences (*e.g.*, invoking an email-sending tool with a fake email address prior to confirming the recipient's address). These results further show that the MCP-R1 can significantly enhance the model's planning capabilities, especially in orchestrating tool-use sequences and maintaining logical integrity during task execution.

## 5 CONCLUSION

In this paper, we introduce MCP-R1, a pipeline that spans from tool construction and data acquisition to training. MCP-R1 is designed to enhance the model's general-purpose tool-use capabilities, enabling it to interact with real-world environments in daily life. Experiments demonstrate that MCP-R1 not only improves performance on search tasks but also enhances the generalization of tool-calling abilities, leading to significant performance improvements in real-world, everyday tasks. We will work on training more versatile and general-purpose agentic models in the future.

## REPRODUCIBILITY STATEMENT

We are fully committed to the reproducibility of the results reported in this paper. Section 3.1 provides a detailed description of the data synthesis pipeline, and Section 4.2 presents the complete implementation details of the evaluation setup. The appendix includes all prompts and tool specifications used in our experiments. We commit to releasing the full training data, training and evaluation code, test benchmarks, and model checkpoints to the community in future updates.

## ETHICAL STATEMENT

All authors of this paper have read and agreed to abide by the ICLR Code of Ethics. Our research focuses on building a scalable data generation pipeline and developing training stages for tool-interaction capabilities. All data and non-public benchmarks used in this paper were generated by our own pipeline. We confirm that the data contains no bias toward any group and does not involve risks related to privacy, safety, or harmful use. During paper preparation, we used large language models solely for correcting spelling and grammatical errors, and did not rely on them for substantive content generation.

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

You are an impartial LLM judge. Your task: decide whether the Response is semantically consistent with the Reference Answer.

Judging Principles (semantic match rules):
- Focus on whether both texts refer to the same entities, facts, and intended meaning.
- Ignore differences in capitalization, wording order, synonyms, plural vs singular, tense, minor formatting, or extra neutral filler.
- Output 1 (match) only if there is no contradiction and no missing critical fact that changes the core meaning.
- Output 0 (mismatch) if there is contradiction, a different entity, a changed key fact (numbers, names, polarity), or the Response is irrelevant.
- If partially correct but a key element is missing or wrong, output 0.
- Do NOT penalize harmless added clarifications that do not alter meaning.

Output format:
1. First, provide a brief reasoning paragraph (in English) explaining your decision.
2. On the LAST line, output ONLY one of:
   <score>1</score>
   or
   <score>0</score>
No other text after the score line.

Now, judge the consistency between the following Reference Answer and Response:
   Question: {question}
   Reference Answer: {reference_answer}
   Response: {response}

Figure 6: Judge prompt of answer-driven tasks.

## APPENDIX

## A  DETAILS OF MCP SERVERS AND TOOLS

In this section, we provide more details about our hybrid MCP tool systems. We first list all the servers and tools in Tab. 4. For virtual tools, we refer to open-source MCP codebases as follows:

- `https://github.com/blazickjp/arxiv-mcp-server`
- `https://github.com/modelcontextprotocol/servers`
- `https://github.com/modelcontextprotocol/servers`
- `https://github.com/MrCare/mcp_tool`
- `https://jina.ai/reader/`

## B  TRAINING DETAILS

We further provide the prompt for reward model during training as shown in Fig. 6 and 7.

## C  DETAILS OF MCP-REALWORLD

We construct MCP-RealWorld, a benchmark comprising 199 real-world tasks from daily life, including various scenarios such as working, traveling, and daily life. Within MCP-RealWorld, we have excluded tasks that can be solved by a single tool call. Consequently, all tasks require the sequential or parallel execution of multiple tools. To evaluate the performance, we utilize gpt-4.1-mini to

Table 4: Server and Tool list of MCP-R1.

| Server | Tool |
|---|---|
| web-search-server | google_search |
| image-describer-server | describe_image |
| fileloader-server | read_file |
| github-server | get_repository_details, list_issues, get_file_content, search_repositories, search_code, search_issues, get_issue |
| weather-server | get_weather, get_weather_forecast |
| code-interpreter-server | execute_code |
| filesystem-server | list_directory, list_directory_tree, read_file, write_file, create_directory, delete_file, delete_directory, move_file, copy_file, search_files, get_file_info, search_within_files |
| translate-server | translate_text, get_supported_languages |
| arxiv-mcp-server | search_papers, download_paper, list_papers, read_paper, get_paper_meta_info |
| flight-ticket-system-server | list_airports, search_flights, book_flight, view_flight_booking, cancel_flight |
| hotel-reservation-server | query_hotels, book_hotel, view_bookings, cancel_booking |
| dining-search-server | query_dining_places |
| meeting-manager | schedule_meeting, adjust_meeting, cancel_meeting, query_schedule, get_meeting_detail |
| google-calendar-server | add_event, get_events, delete_event, modify_event |
| gmail-server | send_email, create_draft, send_draft, delete_draft, list_emails, view_email, search_emails |
| cloud-drive-server | upload, download, list, delete, size |
| read-page-server | read_page |

assess whether the generated trajectory can successfully complete the entire task. We provide some examples in MCP-RealWorld as shown in Fig. 8

## D   LIMITATIONS AND FUTURE DIRECTION

Although we have demonstrated promising results in this work, certain limitations remain: 1) Constrained by computational resources, we have not explored the potential for training larger-scale models via reinforcement learning within the Virtual-Real Integrated MCP Tool System. In future work, we shall seek to extend the existing training paradigm to larger models, thereby fully unlocking the potential for agentic models to improve themselves through interaction with the environment. 2)Unimodality: We have explored only the agent model's capability to perform practical tasks within the pure language domain, and we believe that native multimodal perception capabilities will significantly broaden the scope of tasks agents can execute. We are planning to extend the existing training framework to multimodal scenarios, synthesise application tasks specific to multimodal contexts, and design perception tools suitable for multimodal environments. This will enable the development of modern agentic models with a broader capability boundary. 3) Training Overhead: RL training introduces substantial computational and temporal costs. When tool interactions are incorporated, tool responses and result computations further amplify these expenditures. Such high costs will constrain the model's scalability for learning in more real-world scenarios, diverse tool systems, and more practical tasks. One potential approach involves asynchronous execution of tool invocations and training, maximising training efficiency while accounting for tool responses.

Given a task description and a trajectory of tool calls with their responses, please evaluate whether the execution flow can successfully accomplish the given task.

The trajectory is provided as a series of Steps with both tool calls and their responses. (The tool calls listed in the same step may be in a certain order, but they are all executed in parallel. Do NOT assume the order in which tool calls within a step complete.)
Steps are numbered sequentially (Step1, Step2, ...).
Tool calls within the same Step are executed in parallel; calls in different Steps are executed sequentially.

Your evaluation should consider:

1. Whether the order (Parallel or Sequential) of tool calls matches the logical requirements of the task.
2. Whether the overall trajectory can accomplish the task, even if there are intermediate errors that are later corrected by subsequent tool calls.
3. Whether the selected tools and their combination can successfully complete the task.

Important evaluation guidelines:
1. First, carefully analyze and understand the task intent: Before evaluating the trajectory, clearly identify what the task is asking for and what the desired outcome should be. Ensure trajectory operations match task intent: The operations in the trajectory should directly serve the purpose and objective identified in the task description. Pay special attention to whether there are some judgment conditions in the task.
2. Pay attention to the execution order logic:
- If a task can be completed with either parallel or sequential execution, both approaches are acceptable.
- If a task requires sequential processing (e.g., one tool's output is needed as input for another), but the trajectory shows parallel execution, this should be judged as incorrect.
- Ensure that dependencies between tool calls are properly respected in the execution order.
- Evaluate based on whether the task objective is ultimately achieved with the correct execution logic.
3. If errors occur during the trajectory but are corrected by later tool calls, the trajectory should still be considered successful.
4. Do not be overly strict about minor textual details mentioned in the task (e.g., if the task mentions "top result" but the search returns 10 results, this is acceptable as long as the task can be completed).
5. Check for task relevance:
- All tool calls in the trajectory should be relevant to accomplishing the given task.
- If the trajectory contains operations that are unrelated to the task (e.g., sending emails when the task is to check weather), this should be judged as incorrect.
- Only tool calls that contribute to or support the completion of the specified task should be present.
6. Ensure that conditional business rules are respected, especially for tasks requiring checks before actions:
- If the task requires that an action (such as creating an event, booking a slot, or making a reservation) is only performed if a certain condition is met (e.g., the time slot is free, no conflicts exist), the trajectory must strictly adhere to this logic.
- The trajectory must not alter existing data (e.g., delete or modify conflicting events) to artificially satisfy the condition unless the task explicitly instructs to do so.
- If the trajectory bypasses the intended condition by removing, changing, or ignoring existing items in order to fulfill the action, this should be judged as incorrect.
- Only if the condition is genuinely met (e.g., the slot is truly free without intervention) should the action be performed.
- If the trajectory violates this rule, output <score>0</score>.
7. If the task cannot be completed due to incomplete information provided by the task description, or insufficient data returned from the environment, but the trajectory correctly executes all reasonable steps and checks according to the available information, consider the trajectory successful.
- If the trajectory follows the correct logical process, checks all necessary conditions, and refrains from taking inappropriate actions when required information is missing, it should be judged as successful.
- For example, if a required email or data is missing, and the trajectory halts further actions accordingly, this is considered correct.

For your response:

Clearly state the reason for your judgment, focusing on whether the task can be completed successfully.
Output your final score as <score>1</score> if the trajectory can accomplish the task, or <score>0</score> if it cannot.
Input:

Task description:
{question}
Trajectory of tool calls and responses:
{trajectory}

Output:
Reason(s) for your evaluation and finally score in <score></score>."""

Figure 7: Judge prompt of goal-driven tasks.

I will be attending a business meeting with a valued client, Ms. Yamamoto Yoko, in Dubai on November 21, 2025. Since the meeting is intended to be more private and informal, could you assist me in locating a quiet coffee shop near Al Marmoom Desert Conservation Reserve? Please add the location to my calendar and notify Ms. Yamamoto of the venue.

"You are preparing for an upcoming international business collaboration project and need to schedule a kickoff meeting with key team members from different parts of the world. Specifically, you need to coordinate with Shannon Schwartz from the United States and Mirja Jacobi Jäckel-Fröhlich from Germany. Your objective is to find a suitable meeting time that falls between 10:00 and 20:00 local time for both participants. Once the meeting time is secured, ensure to send them an email notification confirming the details.

I'm currently planning to attend a significant industry conference in the latter part of 2025. Before diving into the conference activities, I want to ensure the project we're launching is off to a solid start. Please find the available time and organize a project kick-off meeting for the afternoon of 2025-11-08? Please create a calendar invitation with an online meeting link and send it to ['James Cabrera (email: raven40@gmail.com)', 'Claire Walker-Pritchard (email: stevenellis@hotmail.com)', 'Céline Fontaine (email: alphonseleduc@wanadoo.fr)']

Figure 8: Examples of tasks in MCP-RealWorld.

