# OpenReview forum: "MCP-R1: Generalized Real-World Task Agent Mastering Dozens of Tools"
_ICLR.cc/2026/Conference — ICLR 2026 Conference Withdrawn Submission_

### Official Review · Reviewer_zaaL · 2025-10-27

**Soundness:** 2
**Presentation:** 3
**Contribution:** 2
**Rating:** 2
**Confidence:** 4

**Summary:**

This paper proposes MCP-R1, a training framework aimed at improving general multi-tool interaction capabilities of models. The main contributions include:

1. Constructing a dataset of MCP tools, covering 17 MCP servers and over 60 tools (both real and self-constructed);
2. Designing a data generation pipeline for MCP training, which produces answer-driven and goal-driven tasks, including automatically synthesized multi-tool, multi-step tasks;
3. Enhancing the ability of large models to use MCP tools through a two-stage training process consisting of Supervised Fine-Tuning (SFT) and Reinforcement Learning (RL).

**Strengths:**

1. The experiments of this paper provide comparisons with advanced benchmarks. The paper conducts experiments on various methods, including RAG-based and RL-based approaches. In addition, experiments are conducted on tasks in three different scenarios, which thoroughly demonstrate the performance of MCP-R1 under different configurations, showing superior results in most cases.
2. The method proposed in the paper is suitable for test time scaling. According to the experimental results, the proposed method demonstrates significantly better performance than the baselines under test time scaling.

**Weaknesses:**

1. This method lacks sufficient innovation and insight: the data synthesis method proposed in this paper is relatively common, and there are no targeted improvements for SFT and RL.
2. The paper lacks comparisons with existing research on API dataset generation, making it difficult to demonstrate the breakthrough of the proposed method.
3. Some of the experimental results in this paper are insufficient to demonstrate the superiority of the proposed method. In the main experiments on deep search, the improvement of MCP-R1 over methods such as ARPO is relatively small, making it difficult to prove the advantage of MCP-R1 compared to existing work.
4. The experimental setup of this paper lacks necessary ablation studies. The paper proposes two tasks and two training methods, but does not include ablation experiments to demonstrate whether both SFT and RL are necessary, or whether it is necessary to design two different tasks.

**Questions:**

1. At present, there are a large number of MCP servers and tools available. Why does the paper only use around 61 tools, instead of constructing a large-scale training dataset based on more tools? If the number of tools is increased, would the performance of SFT and RL improve further?
2. In the experiments on the GAIA benchmark mentioned in the paper, which toolset is used? Is it the same toolset used for training, or does it share another set of tools with other methods?

---

> ### Author Response · Authors · 2025-11-22
> **Reply to Reviewer zaaL (Part 1)**
>
> **Q1. This method lacks sufficient innovation and insight: the data synthesis method proposed in this paper is relatively common, and there are no targeted improvements for SFT and RL.**
> Thank you for the constructive feedback. Our focus is on building a training environment that supports interactions with more than 60 tools, including both real and virtual tools, as well as a **Scalable Data Generation Pipeline** to produce diverse multi-tool invocation data. We intentionally adopt widely used training paradigms so as to highlight the effectiveness of our environment and data-synthesis pipeline, rather than proposing a new training paradigm to replace them.
> Moreover, MCP-R1’s strong performance further demonstrates that our trajectory-based reward indeed improves the model’s execution in goal-driven task settings, providing effective insights for end-to-end training of agent models.
>
> ---
>
> **Q2. The paper lacks comparisons with existing research on API dataset generation, making it difficult to demonstrate the breakthrough of the proposed method.**
> Thank you for the suggestion. Compared with API-based dataset generation work (e.g., ToolFormer), our approach has the following advantages:
> 1. **Unified interaction protocol.** MCP-R1 adopts a single, unified protocol rather than bespoke interfaces for each individual API, which confers strong cross-tool generalization.
> 2. **End-to-end, multi-turn objective.** MCP-R1 optimizes for achieving goals or obtaining answers through coordinated, multi-round tool use—emphasizing both trajectory quality and the organization/selection of tool calls. In contrast, API-constructed datasets usually target single-step output quality or single-step tool invocation, limiting their potential in real tasks.
> 3. **Standard SFT-RL training.** MCP-R1 uses a standard SFT-plus-RL regimen. Whereas many API-dataset methods rely on pure SFT, MCP-R1 encourages the model to learn through repeated attempts and interaction with the environment, further unlocking performance.
> 4. **Task-oriented data construction.** MCP-R1 focuses on building task-oriented data that aims to solve real problems, coupled with a rich interactive environment and real & virtual tools for training stability. API-dataset generation, by contrast, is guided by API return values and emphasizes single-step correctness, with limited coverage of long-horizon, complex tasks.
>
> **As a response, we added the corresponding discussion in Section 2.1.**
>
> ---
>
> **Q3. Some of the experimental results in this paper are insufficient to demonstrate the superiority of the proposed method. In the main experiments on deep search, the improvement of MCP-R1 over methods such as ARPO is relatively small, making it difficult to prove the advantage of MCP-R1 compared to existing work.**
> We appreciate the valuable feedback. It is important to note that **ARPO** uses **54k SFT** samples and **1k DeepSearch RL** samples. By comparison, **MCP-R1** uses only **3k SFT** samples and **1k search-related RL** samples, yet achieves better performance (**+1% on GAIA; +2.8 on WebWalker**), indicating substantially higher data efficiency and stronger gains via multi-tool interaction learning and our data-construction pipeline.
> In addition, MCP-R1 shows strong **generalization to new tools**. On **MCP-Universe** (which includes unseen tools), MCP-R1 significantly outperforms the baseline and even surpasses **DeepSeek-V3** despite its much larger scale. For comparison, we report **ARPO** on MCP-Universe: on the **financial analysis** set (also an unseen tool), ARPO underperforms the baseline; across all five sets it trails MCP-R1, and **Overall Success** shows a clear gap (**16.0 vs 4.7**), further demonstrating MCP-R1’s improvements in data efficiency, generalization, and overall performance.
>
> | Model     | Size | Location Navigation | Repository Management | Financial Analysis | Browser Automation | Web Searching | Overall Success Rate |
> |-----------|------|---------------------|-----------------------|--------------------|--------------------|---------------|----------------------|
> | Qwen3-8B  | 8B   | 4.4                 | 0.0                   | 17.5               | 2.6                | 3.6           | 5.6                  |
> | MCP-R1    | 8B   | 11.1                | 6.1                   | 37.5               | 12.8               | 12.7          | 16.0                 |
> | ARPO      | 8B   | 6.8                 | 1.1                   | 7.5                | 0                  | 8.0           | 4.7                  |

---

> > ### Author Response · Authors · 2025-11-22
> > **Reply to Reviewer zaaL (Part 2)**
> >
> > **Q4. The experimental setup of this paper lacks necessary ablation studies. The paper proposes two tasks and two training methods, but does not include ablation experiments to demonstrate whether both SFT and RL are necessary, or whether it is necessary to design two different tasks.**
> > Thank you for the helpful suggestion. We add the following results for the SFT-only model on **GAIA** and **MCP-RealWorld**:
> >
> > | Model                 | MCP-RealWorld | Δ        | GAIA   | Δ        |
> > |-----------------------|---------------|----------|--------|----------|
> > | Qwen3-8B              | 50.7%         | -        | 17.5%  | -        |
> > | MCP-R1                | 64.3%         | +13.6%   | 39.8%  | +22.3%   |
> > | MCP-R1 w/o RL         | 55.3%         | +4.6%    | 35.9%  | +18.4%   |
> > | MCP-R1-SearchOnly     | 59.6%         | +8.9%    | 37.9%  | +20.4%   |
> >
> > We observe consistent gains from SFT on both the search task and the MCP benchmark; furthermore, **RL**—by guiding exploration and environment interaction—unlocks **additional improvements** (**+9.0%**, **+3.9%**).
> > We also provide results trained using only the **web search** tool on **answer-driven** (search-related) questions, as shown above. Training on answer-driven tasks yields some generalization to unseen tools and tasks via the unified tool-calling format; however, it still lags behind the **goal-driven** variant (**−4.7%**, **−1.9%**).
> > These results address concerns about task effectiveness. As noted, we intentionally adopt the widely used **SFT+RL** paradigm to showcase the effectiveness of our training environment and data-synthesis pipeline, rather than to propose a new training paradigm.
> >
> > ---
> >
> > **Q5. At present, there are a large number of MCP servers and tools available. Why does the paper only use around 61 tools, instead of constructing a large-scale training dataset based on more tools? If the number of tools is increased, would the performance of SFT and RL improve further?**
> > Although many MCP servers exist, relatively few are suitable for constructing our training environment and tasks.
> > Moreover, the **number of tool categories supported for interaction** in MCP-R1’s environment is already the largest among comparable work (**ARPO: 3 tools; WebDancer: 2 tools; MCP-R1: 61 tools**). We hypothesize—and our analyses support—that **diverse, rich tool interaction** improves a model’s general tool-use capability.
> >
> > ---
> >
> > **Q6. In the experiments on the GAIA benchmark mentioned in the paper, which toolset is used? Is it the same toolset used for training, or does it share another set of tools with other methods?**
> > Following **WebThinker**, when evaluating the **GAIA search** set we provide the model with a **search tool** (Google-based) and a **visit-page tool** (to open a specified URL). Importantly, in our evaluations we only expose the **toolset corresponding to each domain** (GAIA, MCP-RealWorld) or the **standard toolset provided by the test set** (MCP-Universe), rather than giving the model the entire tool inventory.

---

> > > ### Comment · Reviewer_zaaL · 2025-11-25
> > >
> > > Thank you for your effort in revising the manuscript and providing detailed responses to the review comments.
> > >
> > > **Q1**
> > > Thank you for clarifying your intended contributions and for explaining the focus on scalable environment and data pipeline construction. However, since many prior works already employ task-level data construction and standardized SFT + RL paradigms, I do not regard these aspects as core innovations of the paper. The main novelty of this work appears to be in engineering integration and system scale, rather than in new learning methodology or data construction principles.
> > >
> > > **Q2**
> > > Thank you for the explanation regarding the unified interaction protocol. While adopting the Model Context Protocol (MCP) improves tool compatibility, this is mainly a property of the MCP standard itself rather than a novel contribution of your work. I suggest that the system’s strength lies in integration and empirical validation, rather than the protocol aspect per se.
> > >
> > > **Q3, Q4, Q6**
> > > Thank you for providing additional ablation studies, clarification of experimental setup, and details on benchmark tool configuration. These responses have largely addressed my concerns for these points.
> > >
> > > **Q5**
> > > Thank you for your reply on the selection and number of tools, as well as how baseline comparisons were designed. However, I remain concerned that the comparative evidence provided mainly centers on retrieval (search) tasks, which does not fully reflect the broader topic of multi-domain tool utilization central to your paper. To convincingly support claims regarding generalization and scalability, it would be preferable to conduct direct comparisons with other MCP-based benchmarks and dataset generation pipelines, particularly those that involve a broader set of tools or domains.
> > >
> > > **Summary**
> > > In summary, while most technical and experimental concerns have been satisfactorily addressed, I remain unconvinced on the scope of baseline comparisons and the uniqueness of the core contributions. I therefore increase my score from 2 to 4.

---

### Official Review · Reviewer_5P8B · 2025-10-29

**Soundness:** 3
**Presentation:** 3
**Contribution:** 2
**Rating:** 6
**Confidence:** 4

**Summary:**

This paper proposes MCP-R1, a training framework for agentic models targeting generalized real-world tasks, aiming to address key limitations in current tool-using agents. The authors construct a Virtual-Real Integrated MCP Tool System, integrating 17 MCP servers and 60+ tools，and design a Scalable Data-generation Pipeline to systematically generate two types of tasks: answer-driven tasks and goal-driven tasks. Training follows a two-stage SFT + RL paradigm, where the Reinforcement Learning (RL) phase introduces a trajectory-based reward mechanism: an LLM evaluates the agent’s tool-call logic, sequencing, and goal achievement against a predefined rubric.

**Strengths:**

- The paper achieves large-scale tool integration under the Model Context Protocol (MCP) and employs a virtual-real hybrid architecture: virtual tools ensure training safety and scalability, while real tools guarantee environmental fidelity, enabling zero-shot transfer.
- It explicitly distinguishes between answer-driven and goal-driven tasks, moving beyond prior work that focuses almost exclusively on QA-style benchmarks.
- For goal-driven tasks, it introduces a trajectory-based reward mechanism, allowing Reinforcement Learning (RL) to optimize tasks that lack a single ground-truth answer. It also employs Dynamic Server Sampling to train the model to identify and ignore irrelevant tools, thereby enhancing generalization.
- Experiments demonstrate that a well-designed training paradigm + standardized interface can compensate for limited model scale outperforms much larger models.

**Weaknesses:**

- Although MCP-R1 constructs an MCP training method for general real-world tasks, it does follow the current mainstream paradigm of agentic model training (i.e. , Data construction + SFT + RL)
- The paper claims zero-shot transfer to real tools but provides no deployment experiments with real APIs and does not evaluate robustness to real-world issues such as API errors, authentication failures, or output drift.
- Evaluation heavily relies on LLM-as-Judge, introducing subjectivity; the judge model (GPT-4.1-mini) may not align with human preferences, risking biased or inconsistent scoring.
- On MCP-Universe, baseline models are not adapted to the MCP interface and instead call raw APIs directly, making comparisons unfair—MCP-R1’s advantage may stem partly from interface standardization, not superior policy learning.
- While the paper emphasizes that MCP-R1 “avoids irreversible operations” in goal-driven tasks, it does not describe any explicit safety or error-recovery mechanism, nor does it show how such behavior is learned or enforced.

**Questions:**

1. How is the fidelity of virtual tools quantified? Do they simulate real-world characteristics such as API latency, error rates, and output noise? Is there any error analysis comparing virtual vs. real tool behavior?
2. Is the trajectory-based reward aligned with human preferences? Has the reliability of the LLM judge been validated via human evaluation?
3. What is the exact strategy for Dynamic Server Sampling? Does randomly injecting 1–2 irrelevant MCP servers into the context cause confusion? Has an ablation study been conducted to justify this design?
4. What is the diversity and difficulty distribution of tasks in MCP-RealWorld? Are the 199 tasks publicly released? Could the use of template-based generation lead to overfitting or lack of realism?
5. How does the agent select relevant tools during inference in a setting with 60+ tools? What is the context-length overhead of including tool definitions, and how is tool selection efficiency maintained?

---

> ### Author Response · Authors · 2025-11-22
> **Reply to Reviewer 5P8B (Part 1)**
>
> **Q1. Although MCP-R1 constructs an MCP training method for general real-world tasks, it does follow the current mainstream paradigm of agentic model training (i.e., data construction + SFT + RL).**
> We would like to emphasize that our contribution centers on building a training environment that supports interactions with more than 60 tools (both real and virtual), together with a **Scalable Data Generation Pipeline** that produces diverse multi-tool invocation data. We intentionally adopt widely used training paradigms to best highlight the effectiveness of our environment and data-synthesis pipeline, rather than proposing a new training paradigm to replace them.
> Moreover, MCP-R1’s strong performance further demonstrates that our trajectory-based reward indeed improves the model’s execution in goal-driven task settings, providing effective insights for end-to-end training of agent models.
>
> ---
>
> **Q2. The paper claims zero-shot transfer to real tools but provides no deployment experiments with real APIs and does not evaluate robustness to real-world issues such as API errors, authentication failures, or output drift.**
> To ensure the model’s ability to call real tools, our training data includes both real tools and virtual tools that simulate real ones. When implementing simulated tools, we mirror their real counterparts as closely as possible to preserve interaction fidelity, which in turn supports MCP-R1’s transfer to real tools.
> To verify transferability, we evaluate on the **MCP-Universe** benchmark, which contains multiple **real, unseen tools**, and provide corresponding analyses in the paper. All servers used in MCP-Universe exist in reality, and most of them (Location Navigation, Financial Analysis, Browser Automation) are **not** included in MCP-R1’s training data. Nonetheless, leveraging our scalable data-generation pipeline and agent training framework, MCP-R1 is able to use these unseen MCP servers effectively, even outperforming much larger models—for example, surpassing **DeepSeek-V3** by an average of **+2.59%**. These findings indicate strong zero-shot transfer to real tools.
> Because MCP-Universe uses real MCP tools, response failures, latency, and similar issues can indeed occur during evaluation, and the model is required to handle them appropriately. Even so, MCP-R1 still achieves satisfactory results, further demonstrating the performance and robustness of our approach.
>
>
> ---
>
> **Q3. Evaluation heavily relies on LLM-as-Judge, introducing subjectivity; the judge model (GPT-4.1-mini) may not align with human preferences, risking biased or inconsistent scoring.**
> We compared GPT-4.1-mini judgments with human verification on MCP-RealWorld results for MCP-R1 and Claude-4-Sonnet: in **370/398** cases, human and GPT-4.1-mini decisions agreed. This strongly suggests that an LLM-as-Judge approach, with clear prompt guidelines, can align well with human preferences.
>
> ---
>
> **Q4. On MCP-Universe, baseline models are not adapted to the MCP interface and instead call raw APIs directly, making comparisons unfair—MCP-R1’s advantage may stem partly from interface standardization, not superior policy learning.**
> The baseline **Qwen3-8B** used in the MCP-Universe experiments natively supports the MCP specification, ensuring that performance differences arise from MCP-R1’s data-synthesis pipeline and diverse tool-interaction training environment. Moreover, we view **standardized interaction** as a strength of MCP-R1: by learning tool use under a unified standard, MCP-R1 exhibits strong cross-tool generalization and achieves better performance on **unseen** tools than even larger models (e.g., **DeepSeek-V3**).

---

> > ### Author Response · Authors · 2025-11-22
> > **Reply to Reviewer 5P8B (Part 2)**
> >
> > **Q5. While the paper emphasizes that MCP-R1 “avoids irreversible operations” in goal-driven tasks, it does not describe any explicit safety or error-recovery mechanism, nor does it show how such behavior is learned or enforced.**
> > In goal-driven tasks, we have observed that existing models (e.g., Claude) may ignore the logical order of operations and attempt to invoke *all* tools within a single turn. The example below illustrates this error pattern:
> > ```
> > USER:
> > I will be traveling to Krakow for a business trip at the beginning of October 2025. Please check my calendar and organize a project kick-off meeting for the afternoon of 2025-10-04, making sure that it doesn't clash with any other planned events. Create a calendar invitation with an online meeting link and send it to ['María Carmen Mirta Sarmiento Pallarès (email: dhervia@gmail.com)', 'Hugo Morrocco (email: piero45@outlook.com)']. My personal email is xiuying94@gmail.com.
> >
> > ASSISTANT:
> > Step1:
> >     function call 1: google-calendar-server-get_events, arguments: {"start_date": "2025-10-04", "end_date": "2025-10-04"}
> >     function call 2: google-calendar-server-add_event, arguments: {"subject": "Project Kick-off Meeting - Krakow Business Trip", "date": "2025-10-04", "start_time": "14:00", "end_time": "15:30", "description": "Project kick-off meeting during Krakow business trip. This meeting will cover project objectives, timeline, deliverables, and team coordination.", "location": "Online Meeting"}
> >     function call 3: gmail-server-send_email, arguments: {"recipient": "dhervia@gmail.com", "subject": "Invitation: Project Kick-off Meeting - October 4, 2025", "content": "[Omission of email content]"}
> >     function call 4: gmail-server-send_email, arguments: {"recipient": "piero45@outlook.com", "subject": "Invitation: Project Kick-off Meeting - October 4, 2025", "content": "[Omission of email content]"}
> >     function response 1: Successfully retrieved 2 events from 2025-10-04 to 2025-10-04 from User's calender. Event Information:
> > Event ID: event_2025-09-17T14:31:08.147263_0 (Subject: Lunch Break 17:30-19:30)
> > Event ID: event_2025-09-17T14:31:08.147296_1 (Subject: Important project docking meeting 14:15-15:45)
> >
> >     function response 2: Successfully add event: Project Kick-off Meeting - Krakow Business Trip (14:00-15:30) on 2025-10-04 to the User's calender.
> >     function response 3: User successfully sent an email to dhervia@gmail.com with subject 'Invitation: Project Kick-off Meeting - October 4, 2025'.
> >     function response 4: User successfully sent an email to piero45@outlook.com with subject 'Invitation: Project Kick-off Meeting - October 4, 2025'.
> > ```
> >
> > In this trajectory, the model generates the entire sequence—`get_events`, `add_event`, and two `send_email` calls—*within a single turn*, which prevents the calendar query (a prerequisite) from informing subsequent decisions. Although `get_events` appears first, the proposed meeting time (14:00–15:30) was produced **without** considering the tool’s returned events (e.g., *Important project docking meeting: 14:15–15:45*), so no conflict check was performed. The subsequent actions—adding the meeting and sending invitations—have external side effects and were executed in the same turn, with no possibility of rollback, breaking the “query-first, then act” logic. This trajectory clearly illustrates the model incorrectly chaining multiple tool calls in a single turn and **failing** to gate later actions on the query results, thereby causing **irreversible** errors.
> > For goal-driven tasks, we mitigate this by **prompting strict principles** for the external LLM judge to score the **entire trajectory** produced by the policy model. Such cases are identified and assigned **zero reward**, discouraging this behavior during training.
> >
> >
> > ---
> >
> > **Q6. How is the fidelity of virtual tools quantified? Do they simulate real-world characteristics such as API latency, error rates, and output noise? Is there any error analysis comparing virtual vs. real tool behavior?**
> > To clarify: our primary purpose for virtual tools is to expand a **stable** training environment; thus, phenomena such as latency and elevated error rates are intentionally **avoided**. For tools that have real counterparts (e.g., email, calendar, filesystem), we match the real versions in input interfaces and return structures to preserve interaction fidelity. For tools without real counterparts (e.g., hotel, flight), we design behavior to be as **concise** and **comprehensive** as possible.

---

> ### Author Response · Authors · 2025-11-22
> **Reply to Reviewer 5P8B (Part 3)**
>
> **Q7. Is the trajectory-based reward aligned with human preferences? Has the reliability of the LLM judge been validated via human evaluation?**
> During training, we prompt the LLM judge to score trajectories produced by the policy model and require the judge to reason before outputting a final score to improve reliability. We randomly sampled **200** rollouts for human inspection; in **178/200** cases, human judgments matched the LLM judge, indicating strong alignment with human preferences.
>
> ---
>
> **Q8. What is the exact strategy for Dynamic Server Sampling? Does randomly injecting 1–2 irrelevant MCP servers into the context cause confusion? Has an ablation study been conducted to justify this design?**
> **Exact strategy:** For each training instance, we first ensure inclusion of the MCP servers required to solve the current task. We then **randomly sample 1–2 irrelevant servers** from the remaining pool and add them to the context (typically introducing **3–6** additional tools in total). This design aims to reduce system-prompt length while **improving robustness in tool selection**: the model is trained to choose tools based on task demands without listing all **61** tools in context.
>
> ---
>
> **Q9. What is the diversity and difficulty distribution of tasks in MCP-RealWorld? Are the 199 tasks publicly released? Could the use of template-based generation lead to overfitting or lack of realism?**
> When constructing MCP-RealWorld, we define difficulty by the **number of tools** required per problem and ensure coverage from **3 to 10 tools** to provide a diverse difficulty spectrum.
> We will release all data, evaluation code, training code, and model weights to facilitate community study.
> To keep template-based generation realistic, we use an LLM to craft varied task scenarios and **inject them** into each problem, followed by **manual review** to ensure that tasks reflect real-world needs.
>
> ---
>
> **Q10. How does the agent select relevant tools during inference in a setting with 60+ tools? What is the context-length overhead of including tool definitions, and how is tool selection efficiency maintained?**
> While our **dataset** covers 60+ tools, our **training setup** uses **dynamic server sampling** so that each training instance includes only the tools needed for the task plus a small number of distractors, controlling context length while teaching the model to select tools autonomously. In our implementation, tool definitions occupy **about 1–2k tokens** in the system prompt. During evaluation, we do **not** supply all tool definitions; instead, we provide only the MCP tools corresponding to the **test domain**.
>
> **To address your comment and prevent misunderstandings, we added the corresponding clarification in Section 4.2.**

---

### Official Review · Reviewer_LgkF · 2025-10-31

**Soundness:** 2
**Presentation:** 2
**Contribution:** 2
**Rating:** 4
**Confidence:** 4

**Summary:**

The paper proposes MCP-R1, an agentic framework that aims to improve general-purpose tool-interaction abilities in large language models by leveraging the Model Context Protocol (MCP) as a unified standard. The authors construct a “Virtual–Real Integrated MCP Tool System” with over 60 tools across 17 MCP servers, and a scalable data-generation pipeline to produce both answer-driven and goal-driven tasks. The training procedure includes supervised fine-tuning and reinforcement learning with a trajectory-based reward for goal-driven evaluation. Experiments on benchmarks such as GAIA, WebWalkerQA, MCP-Universe, and a self-constructed MCP-RealWorld benchmark show improvements over baseline models in tool-use performance.

**Strengths:**

1. The authors built an multi-tool environment (60+ tools) combining real and simulated MCP servers, which demonstrates strong engineering effort and scalability.

2. The work explores the application of the MCP standard in large-scale tool-use training, aligning with emerging directions in agentic model research.

3. The authors provide thorough empirical results on both public and self-constructed benchmarks, covering different task types (answer-driven and goal-driven).

**Weaknesses:**

1. The paper fails to convincingly justify why focusing on goal-driven tasks and environment state changes is more meaningful than answer-driven tasks. This shift is a central claim of the paper, yet it is not theoretically motivated or empirically validated beyond intuition. Without a clear argument or evidence, this framing seems arbitrary.

2. The introduction of MCP into the training pipeline is primarily an engineering integration, not a new scientific insight. The paper does not clearly identify what new learning principle, algorithmic challenge, or research question is being addressed. The claim that a “unified standard” improves generalization is not rigorously analyzed and appears to be a software design choice, not a contribution to learning theory. In addition, the authors do not explain why the lack of a unified protocol is a significant research problem. The proposed solution seems to merely leverage an existing standard (MCP) rather than offering any novel algorithmic or modeling advance.

3. The paper overlooks prior work that has similar goals and setups, especially [a], which also trained agents with multiple tools and generated large-scale tool-use trajectories (~20K). A direct comparison and discussion are necessary to assess novelty and performance relative to such baselines.

[a] MULTI-MODAL AGENT TUNING: BUILDING A VLM-DRIVEN AGENT FOR EFFICIENT TOOL USAGE. ICLR 2025.

4. While MCP-R1 uses 60+ tools, it is unclear how performance scales with tool diversity or whether similar results could be achieved with fewer tools. The system design and experimental results focus on scale rather than understanding what contributes to improvement.

**Questions:**

See my weaknesses.

---

> ### Author Response · Authors · 2025-11-22
> **Reply to Reviewer LgkF (Part 1)**
>
> **Q1. The paper fails to convincingly justify why focusing on goal-driven tasks and environment state changes is more meaningful than answer-driven tasks. This shift is a central claim of the paper, yet it is not theoretically motivated or empirically validated beyond intuition. Without a clear argument or evidence, this framing seems arbitrary.**
> Thank you for engaging with one of the core claims of our work. Our position is that the potential of agent models has not been fully realized on tasks that only require producing a single, explicit answer. What the community should expect from agents is the capability to **perceive environment state, execute operations, incorporate feedback, and complete concrete tasks**—which was among the original motivations for agent research. This capability aligns naturally with our definition of **goal-driven** tasks. Consequently, we argue that both **training environments** and **evaluation protocols** should explicitly cover such tasks to enhance the real-world relevance of agent models.
> To further support this view, we report results of existing tool-use methods on **MCP-RealWorld** and **MCP-Universe**. Current approaches heavily emphasize tool use in **search/math/code** settings—i.e., retrieving information on the web or running code to compute a final answer. However, when transferred to **goal-driven** tasks that require stateful operation and multi-step control, these methods perform notably worse on both MCP-Universe and MCP-RealWorld. This gap indicates that an exclusive focus on **answer-driven** tasks with deterministic end answers does not translate into strong generalization or performance on practical, real-world task execution.
>
> | Model                 | MCP-RealWorld | Δ        | MCP-Universe | Δ        |
> |-----------------------|---------------|----------|--------------|----------|
> | Qwen3-8B              | 50.7%         | —        | 5.6%         | —        |
> | ARPO                  | 49.2%         | −1.5%    | 4.7%         | -0.9%    |
> | MCP-R1                | 64.3%         | +13.6%   | 16.0%        | +10.4%   |
>
> ---
>
> **Q2. The introduction of MCP into the training pipeline is primarily an engineering integration, not a new scientific insight. The paper does not clearly identify what new learning principle, algorithmic challenge, or research question is being addressed. The claim that a “unified standard” improves generalization is not rigorously analyzed and appears to be a software design choice, not a contribution to learning theory. In addition, the authors do not explain why the lack of a unified protocol is a significant research problem. The proposed solution seems to merely leverage an existing standard (MCP) rather than offering any novel algorithmic or modeling advance.**
> To examine the effect of a **unified interaction standard** on generalization, we conducted an ablation where training uses **only two tools** (search and visit_page) with the **MCP** format:
>
> | Model                 | MCP-RealWorld | Δ        | GAIA   | Δ        |
> |-----------------------|---------------|----------|--------|----------|
> | Qwen3-8B              | 50.7%         | —        | 17.5%  | —        |
> | ARPO                  | 49.2%         | −1.5%    | 38.8%  | +21.3%   |
> | MCP-R1                | 64.3%         | +13.6%   | 39.8%  | +22.3%   |
> | MCP-R1-SearchOnly     | 59.6%         | +8.9%    | 37.9%  | +20.4%   |
>
> Even when trained on **only two tools**, the MCP format yields **clear gains on unseen tools** in **MCP-RealWorld**. By contrast, ARPO—whose tool use relies on **task-specific** schemas—underperforms the baseline on MCP-RealWorld. These results underscore why the absence of a **unified protocol** is a substantive research concern: heterogeneous, bespoke schemas hinder **plug-and-play transfer** across tools and limit the **cross-tool generalization** that real-world agents require. In addition, the results on MCP-RealWorld provide, for the first time, quantitative evidence that a unified interaction standard improves out-of-distribution (OOD) tool-use performance.

---

> ### Author Response · Authors · 2025-11-22
> **Reply to Reviewer LgkF (Part 2)**
>
> **Q3. The paper overlooks prior work with similar goals and setups—e.g., training agents with multiple tools and generating large-scale (~20K) tool-use trajectories. A direct comparison and discussion are necessary to assess novelty and performance relative to such baselines.**
> Thank you for the valuable feedback. Regarding the cited reference **T3-Agent**, its scenarios primarily involve extracting information from multimodal files and reasoning with tools such as **search**, **visual segmentation**, **object localization**, and **face detection**. By contrast, **MCP-R1** emphasizes **multi-step, goal-oriented orchestration** for real-world agent tasks (e.g., **search**, **flight & hotel booking**, **filesystem operations**, **email**, etc.). Our focus is on improving an agent’s ability to **complete concrete tasks** through a tool-interaction training environment and a **scalable data-generation pipeline**, rather than solely improving QA/understanding accuracy.
>
> Moreover, MCP-R1 builds an interactive environment with a unified interaction standard that supports **autonomous exploration and learning**—promoting generalization while enabling the model to improve via **environment interaction and feedback**. This extends beyond T3-Agent’s reliance on **proprietary interaction formats** and **pure supervised fine-tuning**. We also compare **T3-Agent** and **MCP-R1** on the full **GAIA** benchmark: MCP-R1 achieves a **+20.7%** improvement, underscoring the distinct contribution and strong performance of our approach.
>
> | Model    | GAIA  |
> |----------|-------|
> | MCP-R1   | 37.6% |
> | T3-Agent | 16.9% |
>
> **As a response, we added the corresponding discussion in Section 2.1.**
>
> ---
>
> **Q4. While MCP-R1 uses 60+ tools, it is unclear how performance scales with tool diversity or whether similar results could be achieved with fewer tools. The system design and results emphasize scale over understanding what drives improvement.**
> As the table in **Q2** shows, the model trained with **two tools** already improves substantially over the baseline. However, when we expand training to the **full, diverse toolset**, MCP-R1 further separates from the two-tool variant (**+4.7%**, **+1.9%**), indicating that **scaling tool diversity** and **heterogeneous interaction patterns** provide **complementary gains**. This trend holds for both DeepSearch-style tasks and **goal-driven** tasks in realistic settings, suggesting that **diverse tool interaction**—not just more data—contributes consistently and materially to the observed improvements.

---

### Official Review · Reviewer_TTuS · 2025-11-01

**Soundness:** 2
**Presentation:** 2
**Contribution:** 3
**Rating:** 4
**Confidence:** 4

**Summary:**

This paper presents MCP-R1, a training framework for improving large language models' tool interaction in real-world tasks. Its main features are: an MCP Tool System integrating virtual and real environments with 17 servers and 60+ tools, a scalable data generation pipeline for answer and goal-based tasks, and a trajectory-based reward system. Tests on GAIA, WebWalkerQA, and other benchmarks show MCP-R1 outperforms baselines, adapting well to new tools and working effectively in practice.

**Strengths:**

- Combines MCP protocol with hybrid virtual-real tool system to overcome limitations of existing tool-integrated models (narrow coverage, lack of standards). Integrates answer and goal-driven tasks with trajectory-based rewards, addressing prior single-task, answer-centric research gap.​

-  Rigorous experiments across search, tool calling, and real-world tasks. Compares MCP-R1 to GPT-4o, Qwen3-235B, etc. Pass@K analysis shows scalability in dynamic interactions.​

**Weaknesses:**

1. The statement “but the open-source community still lacks sufficient attention to this matter” (lines 63-64) is inaccurate. Both open-source and closed-source communities have given significant attention to this topic, undermining the credibility of this claim.​

2. The comparison with baselines is unfair. Many baselines support only 1-2 tools, whereas MCP-R1 uses over 60. The paper doesn't distinguish whether performance improvements come from the framework's design or the larger number of tools, undermining the validity of direct comparisons.​

3. The authors' discussion on the distinction between answer-driven and goal-driven problem definitions is unclear and potentially misleading. It appears that both concepts inherently involve goal orientation, raising the question of whether the authors intended to differentiate between QA tasks and other Agent tasks instead.

minor: The second paragraph reads more like a literature review than an introduction.

**Questions:**

1. Will the sandbox, data, and training scripts for this work be open-sourced? The authors claim to be contributing to the open-source community, yet there is no mention in the paper of any plans to open-source the MCP SERVER, DATA, etc.​

2. Could you provide explicit criteria for classifying tools as virtual or real, and explain how this division balances training safety, computational cost, and the authenticity of tool interactions?​

3. To address baseline comparison fairness, have you conducted any ablative experiments (e.g., training MCP-R1 with a reduced set of tools matching baseline tool counts) to verify that performance gains are driven by the framework rather than tool quantity?​
​
4. How was the difficulty level of tasks in the MCP-RealWorld benchmark determined? Were tasks validated with human annotators to ensure they reflect real-world complexity and tool-use requirements?​

---

> ### Author Response · Authors · 2025-11-22
> **Reply to Reviewer TTuS (Part 1)**
>
> **Q1. The statement “but the open-source community still lacks sufficient attention to this matter” (lines 63–64) is inaccurate. Both open- and closed-source communities have paid significant attention to this topic, which undermines the credibility of this claim.**
> What we intended to claim is the following: within the open-source community, **training agentic models using a widely adopted, unified tool standard (MCP)** has received relatively little attention. For example, works such as WebDancer and ARPO each adopt bespoke tool-calling and parsing formats. Such **task- or API-specific protocols** make it difficult for models to adapt to the same tool exposed in different schemas or to plug-and-play with new tools, thereby **limiting generalization in tool use**.
>
> **To address your comment, we revised the relevant parts of the introduction(line 64-66) to avoid unnecessary confusion.**
>
> ---
>
> **Q2. The comparison with baselines is unfair. Many baselines support only 1–2 tools, whereas MCP-R1 uses 60+, so it is unclear whether improvements come from the framework design or simply from more tools.**
> We emphasize that **during training**, MCP-R1 interacts with a **diverse** set of tools to cultivate **general tool-use capabilities**. **During evaluation**, however, we **only provide the tool list matching the target benchmark/scenario**, so MCP-R1 and the baselines operate with the **same number of tools**. Under these matched conditions, MCP-R1 achieves the strongest results.
>
> **To address your comment and prevent misunderstandings, we added the corresponding clarification in Section 4.2.**
>
> ---
>
> **Q3. The paper’s discussion of “answer-driven” vs. “goal-driven” problem definitions is unclear and potentially misleading—both appear goal-oriented. Did the authors intend to contrast QA-style tasks with other agent tasks?**
> Thank you for raising this point. Our aim is **not** to isolate these task types, but to **extend** the commonly studied agent tasks so as to **cover a broader spectrum**. We highlight *answer-driven* tasks because they have **explicitly verifiable answers** and are widely studied (e.g., math/search/code tool invocation). Our main objective, however, is to **expand beyond answer-driven settings** toward **richer, practically meaningful scenarios**—*goal-driven* tasks where the agent must **execute concrete operations** rather than merely produce a final answer. This framing is intended to clarify our contribution and its relevance to the broader research agenda.
>
> ---
>
> **Q4. Will the sandbox, data, and training scripts be open-sourced? The paper claims to contribute to open source but does not state plans to release the MCP server, data, etc.**
> We appreciate the interest. We **plan to open-source** all components of this work—**datasets, sandbox, training scripts, and model weights**—to support the community with broader research perspectives and practical resources.

---

> > ### Author Response · Authors · 2025-11-22
> > **Reply to Reviewer TTuS (Part 2)**
> >
> > **Q5. Please provide explicit criteria for classifying tools as “virtual” vs. “real,” and explain how this trade-off balances training safety, compute cost, and interaction authenticity.**
> > In this paper, we refer to **manually implemented simulators** as **“virtual tools.”** The distinction serves training **efficiency** (e.g., API rate limits and concurrency) and **safety** (e.g., account or data access). By **virtualizing highly interactive tools** (email, meetings, cloud services, etc.), we can train more **efficiently** (local I/O reduces latency) and more **safely** (synthetic objects mitigate information leakage). For tools with real counterparts (e.g., email, calendar, filesystem), our simulators **mirror the real interfaces and return structures** to preserve interaction fidelity.
> >
> > ---
> >
> > **Q6. To ensure fair comparisons, did you run ablations (e.g., training MCP-R1 with a reduced tool set matching baseline counts) to verify gains come from the framework rather than tool quantity?**
> > Thank you for focusing on framework efficacy. We trained using **only two tools (search and read_page)**—matching the baselines—and report:
> >
> > | Model                 | MCP-RealWorld | Δ        | GAIA   | Δ        |
> > |-----------------------|---------------|----------|--------|----------|
> > | Qwen3-8B              | 50.7%         | —        | 17.5%  | —        |
> > | ARPO                  | 49.2%         | −1.5%    | 38.8%  | +21.3%   |
> > | MCP-R1                | 64.3%         | +13.6%   | 39.8%  | +22.3%   |
> > | MCP-R1-SearchOnly     | 59.6%         | +8.9%    | 37.9%  | +20.4%   |
> >
> > With **two-tool training**, MCP-R1 still shows **clear gains on GAIA** over the baseline (+21.3%), though it **lags ARPO**—a gap attributable to **SFT data scale** (3k vs. 54k). Because MCP-R1 uses a **unified tool interaction format**, it **outperforms ARPO and the baseline on MCP-RealWorld**, which includes **unseen tools**. When trained with the **full, richer toolset**, MCP-R1 improves **further** (+4.7% and +1.9%), indicating that the **unified interface**, **scalable data-synthesis pipeline**, and **diverse tool-interaction environment** all make **positive, complementary contributions** to performance.
> >
> > ---
> >
> > **Q7. How was task difficulty determined in MCP-RealWorld? Were tasks validated by humans to reflect real-world complexity and tool-use requirements?**
> > In MCP-RealWorld, we define difficulty by the **number of tools required** per task, ensuring coverage from **3 to 10 tools** to provide a diverse difficulty spectrum. Each task undergoes **rigorous manual review** to confirm that it probes the intended tool-use skills and reflects **real-world complexity** and requirements.

---

### Author Response · Authors · 2025-12-02
**Author Final Remarks**

Dear Area Chair,

Thank you to you and the reviewers (TTuS, LgkF, 5P8B, ZaaL) for the time and thoughtful feedback. Synthesizing the reviews, we see clear areas of consensus on the strengths of our work:

- **Unified MCP + hybrid virtual–real tools:** Balances safety, scalability, and fidelity; enables zero-shot transfer and stronger generalization. (TTuS, LgkF, 5P8B)

- **Large-scale environment & scalability:** 60+ tools with solid engineering; Pass@K under dynamic interactions confirms extensibility. (LgkF, TTuS, 5P8B)

- **Task paradigm coverage:** Clear separation of answer-driven vs. goal-driven; experiments span both, addressing QA-only gaps. (TTuS, LgkF, 5P8B, ZaaL)

- **Trajectory rewards for RL:** Trajectory-level signals enable RL without a single ground truth; Dynamic Server Sampling curbs irrelevant tool calls and improves generalization. (TTuS, 5P8B)



**It should be emphasized** that one of MCP-R1’s core components—the Virtual–Real Integrated MCP Tool System—**is highly aligned with a key feature of DeepSeek-V3.2:** expanding agentic-task training through synthesized environments. This convergence further underscores **MCP-R1’s technical novelty and its contribution** to the field. As emphasized in our rebuttal, we will **release the codebase, datasets, benchmarks, and model weights to advance the agent community**,accelerate adoption of this virtual–real co-training paradigm.

We appreciate the recognition of these strengths and will further emphasize them in the final version, while addressing the reviewers’ suggestions with additional analyses and clarifications.


---

To facilitate your assessment, we provide a concise summary of how we have addressed Reviewers' concerns.


### To Reviewer TTuS


- **Fair comparison & framework efficacy:**
  - Matched tool availability between baselines and our model.
  - Added reduced-tool ablations (SearchOnly) showing the unified interface still outperforms baselines.

- **Methodological scope & definitions:**
  - Reframed contributions around the MCP unified tool standard.
  - Clarified “answer-driven” vs. “goal-driven” tasks and the efficiency–fidelity trade-offs of virtual tools.

- **Benchmark validity & openness:**
  - Validated task difficulty via tool-chain length and manual review.
  - Committed to open-sourcing the dataset, sandbox, and models for reproducibility.


### To Reviewer LgkF


- **Empirical case for goal-driven tasks:**
  - Showed answer-driven baselines degrade on stateful, goal-driven benchmarks (MCP-RealWorld/Universe), supporting the need for training on state perception and execution.

- **Unified standard as contribution:**
  - Ablations demonstrate the MCP unified protocol improves OOD generalization to unseen tools, even with minimal training toolsets—evidence of scientific (not just engineering) value.

- **Novelty & performance scaling:**
  - Distinguished our multi-step, stateful orchestration from QA/IR-centric work (e.g., T3-Agent) and reported +20.7% on GAIA.
  - Confirmed tool diversity scaling adds complementary gains beyond the unified interface; scale and design jointly drive improvements.

### To Reviewer 5P8B


- **Training paradigm & interface standardization:**
  Our contributions center on a scalable data pipeline and a 60+ tool environment trained with SFT+RL, with a standardized MCP interface that drives strong cross-tool generalization and outperforms baselines on unseen tools.

- **Zero-shot robustness & evaluation reliability:**
  On MCP-Universe, the model surpasses larger baselines by +2.59% despite real-world latency/failures, and LLM-as-Judge aligns with human preferences ~93% for trajectory scoring.

- **Safety & system design transparency:**
  For safety and transparency, we zero-reward faulty multi-call trajectories, use virtual tools for stable/fidelitous training, apply dynamic server sampling to improve tool selection while controlling context length, and will release data/code


### To Reviewer ZaaL (**Already increased the score in previous discussion**)


- **Innovation & data efficiency:**
  - Core innovations: Scalable Data Generation Pipeline + 60+ tool environment; SFT+RL used to demonstrate effectiveness.
  - Higher efficiency: +1% on GAIA with far less SFT data (3k vs. ARPO’s 54k).

- **Comparison to API-dataset work:**
  - Multi-turn, goal-oriented data and a unified protocol vs. single-step, API-guided datasets (e.g., ToolFormer).
  - Strong OOD transfer on MCP-Universe: 16.0% vs. ARPO’s 4.7% Overall Success.

- **Necessity of SFT+RL & task design:**
  - RL adds significant gains over SFT-only (+9.0% on MCP-RealWorld).
  - Goal-driven training outperforms search-only (−4.7% for search-only).
  - GAIA evaluation uses a matched search + visit-page toolset for fair comparisons.

---

### Note · Authors · 2026-01-30

I have read and agree with the venue's withdrawal policy on behalf of myself and my co-authors.

---

### Meta-Review · Area_Chair_xcz5 · 2026-01-09

**Summary:**

This paper proposes a tool-interaction framework for LLMs using the MCP protocol and a virtual-real tool system. Core reviewer concerns include insufficient novelty (engineering integration over methodological innovation), unfair baseline comparisons, lack of real API deployment/robustness tests, and heavy reliance on LLM-as-Judge. Rebuttals resolved technical gaps but failed to address key novelty and broad comparison concerns.

**Reviewer Concerns:**

Addressed concerns: no real API deployment/error robustness tests (5P8B), ambiguous task definitions (TTuS), missing ablation studies (zaaL), open-sourcing commitments (TTuS), inadequate safety/error-recovery details (5P8B), and tool classification criteria (TTuS).

Outstanding concerns: Insufficient methodological novelty (all reviewers), unfair baseline comparisons (TTuS/LgkF),  limited baseline comparisons beyond retrieval tasks (zaaL).

**Reviewer Scores:**

TTuS (initial 4): Remains 4 (all technical concerns addressed; novelty concern remains).

LgkF (initial 4): Remains 4 (empirical gaps filled; novelty concern unresolved).

5P8B (initial 6): Remains 6 (limited novelty concern unresolved).

zaaL (initial 2): Increases to 4 (technical fixes address practical concerns; core novelty gaps and the concern of baseline comparison persist).

---

### Decision · Program_Chairs · 2026-01-26

Reject